# Lifelong Graph Learning

Chen Wang[†]
chenwang@dr.com

Yuheng Qiu[†]
yuhengq@andrew.cmu.edu

Dasong Gao[†]
dasongg@andrew.cmu.edu

Sebastian Scherer[†]
basti@cmu.edu

[†]The Robotics Institute, Carnegie Mellon University, Pittsburgh, PA 15213

## Abstract

*Graph neural networks (GNNs) are powerful models for many graph-structured tasks. Existing models often assume that a complete structure of a graph is available during training. In practice, however, graph-structured data is usually formed in a streaming fashion so that learning a graph continuously is often necessary. In this paper, we aim to bridge GNN to lifelong learning by converting a graph problem to a regular learning problem, so that GNN can inherit the lifelong learning techniques developed for convolutional neural networks (CNNs). To this end, we propose a new graph topology based on feature cross-correlation, namely, the feature graph. It takes features as new nodes and turns nodes into independent graphs. This successfully converts the original problem of node classification to graph classification, in which the increasing nodes are turned into independent training samples. In the experiments, we demonstrate the efficiency and effectiveness of feature graph networks (FGN) by continuously learning a sequence of classical graph datasets. We also show that FGN achieves superior performance in two applications, i.e., lifelong human action recognition with wearable devices and feature matching. To the best of our knowledge, FGN is the first work to bridge graph learning to lifelong learning via a novel graph topology. Source code is available at* https://github.com/wang-chen/LGL.

## 1. Introduction

Graph neural networks (GNN) have received increasing attention and proved useful for many tasks with graph-structured data, such as citation, social, and protein networks [52]. However, graph data is sometimes formed in a streaming fashion and real-world datasets are continuously evolving over time, thus learning a streaming graph is expected in many cases [46]. For example, in a social network, the number of users often grows over time and we expect that the model can learn continuously with new users. In this paper,

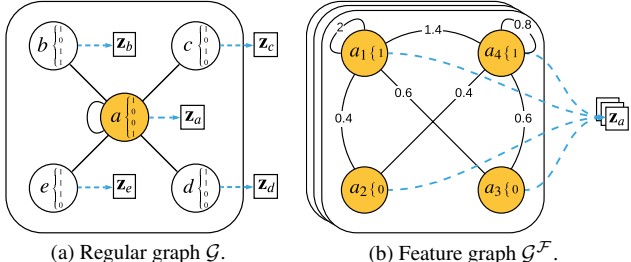

(a) Regular graph $\mathcal{G}$.      (b) Feature graph $\mathcal{G}^{\mathcal{F}}$.

Figure 1. We introduce feature graph network (FGN) for lifelong graph learning. A feature graph takes the features as nodes and turns nodes into graphs, resulting in a graph predictor instead of the node predictor. This makes the lifelong learning techniques for CNN applicable to GNN, as the new nodes in a regular graph become individual training samples. Take the node $a$ with label $\mathbf{z}_a$ in the regular graph $\mathcal{G}$ as an example, its features $\mathbf{x}_a = [1, 0, 0, 1]$ are nodes $\{a_1, a_2, a_3, a_4\}$ in feature graph $\mathcal{G}_a^{\mathcal{F}}$. The feature adjacency is established via feature cross-correlation between $a$ and its neighbors $\mathcal{N}(a) = \{a, b, c, d, e\}$ to model feature "interaction."

we extend graph neural networks to lifelong learning, which is also known as continual or incremental learning [26].

Lifelong learning often suffers from "catastrophic forgetting" if the models are simply updated with new samples [35]. Although some strategies have been developed to alleviate the forgetting problem for convolutional neural networks (CNN), they are still difficult for graph networks. This is because in the lifelong learning setting, the graph size can increase over time and we have to drop off old data or samples to learn new knowledge. However, the existing graph model cannot directly overcome this difficulty. For example, graph convolutional networks (GCN) require the entire graph for training [20]. SAINT [58] requires pre-processing for the entire dataset. Sampling strategies [7, 13, 58] easily forget old knowledge when learning new knowledge.

Recall that regular CNNs are trained in a mini-batch manner where the model can take samples as independent inputs [23]. Our question is: can we convert a graph task into a traditional CNN-like classification problem, so that (I) nodes can be predicted independently and (II) the lifelong learning

techniques developed for CNN can be easily adopted for GNN? This is not straightforward as node connections cannot be modeled by a regular CNN-like classification model. To solve this problem, we propose to construct a new graph topology, the feature graph in Figure 1, to bridge GNN to lifelong learning. It takes features as nodes and turns nodes into graphs. This converts node classification to graph classification where the node increments become independent training samples, enabling natural mini-batch training.

The contribution of this paper includes: (1) We introduce a novel graph topology, *i.e.* feature graph, to convert a problem of growing graph to an increasing number of training samples, which makes existing lifelong learning techniques developed for CNNs applicable to GNNs. (2) We take the cross-correlation of neighbor features as the feature adjacency matrix, which explicitly models feature "interaction", that is crucial for many graph-structured tasks. (3) Feature graph is of constant computational complexity with the increased learning tasks. We demonstrate its efficiency and effectiveness by applying it to classical graph datasets. (4) We also demonstrate its superiority in two applications, *i.e.* distributed human action recognition based on subgraph classification and feature matching based on edge classification.

## 2. Related Work

### 2.1. Lifelong Learning

**Non-rehearsal Methods** Lifelong learning methods in this category do not preserve any old data. To alleviate the forgetting problem, progressive neural networks [36] leveraged prior knowledge via lateral connections to previously learned features. Learning without forgetting (LwF) [24] introduced a knowledge distillation loss [15] to neural networks, which encouraged the network output for new classes to be close to the original outputs. Distillation loss was also applied to learning object detectors incrementally [41]. Learning without memorizing (LwM) [10] extended LwF by adding an attention distillation term based on attention maps for retaining information of the old classes.

EWC [21] remembered old tasks by slowing down learning on important weights. RWalk [6] generalized EWC and improved weight consolidation by adding a KL-divergence-based regularization. Memory aware synapses (MAS) [1] computed an importance value for each parameter in an unsupervised manner based on the sensitivity of output function to parameter changes. [48] presented an embedding framework for dynamic attributed network based on parameter regularization. A sparse writing protocol is introduced to a memory module [43], ensuring that only a few memory spaces is affected during training.

**Rehearsal Methods** Rehearsal lifelong learning methods can be roughly divided into rehearsal with synthetic data or rehearsal with exemplars from old data [33]. To ensure that the loss of exemplars does not increase, gradient episodic memory (GEM) [26] introduced orientation constraints during gradient updates. Inspired by GEM, [2] selected exemplars with a maximal cosine similarity of the gradient orientation. iCaRL [32] preserved a subset of images with a herding algorithm [49] and included the subset when updating the network for new classes. EEIL [5] extended iCaRL by learning the classifier in an end-to-end manner. [51] further extended iCaRL by updating the model with class-balanced exemplars. Similarly, [3, 16] further added constraints to the loss function to mitigate the effect of imbalance. To reduce the memory consumption of exemplars, [18] applied the distillation loss to feature space without having to access to the corresponding images. Rehearsal approaches with synthetic data based on generative adversary networks (GAN) were used to reduce the dependence on old data [14, 40, 50, 53].

### 2.2. Graph Neural Networks

Graph neural networks have been widely used to solve problems with graph-structured data [60]. The spectral network extended convolution to graph problems [4]. Graph convolutional network (GCN) [20] alleviated over-fitting on local neighborhoods via the Chebyshev expansion. To identify the importance of neighborhood features, graph attention network (GAT) [42] added an attention mechanism into GCN, further improving the performance on citation networks and the protein-protein interaction dataset.

GCN and its variants require the entire graph during training, thus they cannot scale to large graphs. To solve this problem and train GNN with mini-batches, a sampling method, SAGE [13] is introduced to learn a function to generate node embedding by sampling and aggregating neighborhood features. JK-Net [54] followed the same sampling strategy and demonstrated a significant accuracy improvement on GCN with jumping connections. DiffPool [57] learned a differentiable soft cluster assignment to map nodes to a set of clusters, which then formed a coarsened input for the next layer. Ying *et al.* [56] designed a training strategy that relied on harder-and-harder training examples to improve the robustness and convergence speed of the model.

FastGCN [7] applied importance sampling to reduce variance and perform node sampling for each layer independently, resulting in a constant sample size in all layers. [17] sampled lower layer conditioned on the top one ensuring higher accuracy and fixed-size sampling. Subgraph sampling techniques were also developed to reduce memory consumption. [8] sampled a block of nodes in a dense subgraph identified by a clustering algorithm and restricted the neighborhood search within the subgraph. SAINT [58] constructed mini-batches by sampling the training graph.

Nevertheless, most of the sampling techniques still require a pre-processing of the entire graph to determine the sampling process or require a complete graph structure,

which makes those algorithms not directly applicable to lifelong learning. In this paper, we hypothesize that a different graph structure is required for lifelong learning, and the new structure is not necessary to maintain its original meaning. We noticed that there is some recent work focusing on continuously learning a graph problem, but they have different formulation. For example, several exemplar selection methods are tested in ER-GNN [59]. A weight preserving method is introduced to growing graph [25]. A combined strategy of regularization and data rehearsal is introduced to streaming graphs in [46]. To overcome the incomplete structure, [12] learns a temporal graph in sliding window.

## 3. Problem Formulation

We start by defining regular graph learning before lifelong graph learning for completeness. An attribute graph is defined as $\mathcal{G} = (\mathcal{V}, \mathcal{E})$, where $\mathcal{V}$ is the set of nodes and $\mathcal{E} \subseteq \left\{ \{v_a, v_b\} \, | \, (v_a, v_b) \in \mathcal{V}^2 \right\}$ is the set of edges. Each node $v \in \mathcal{V}$ is associated with a target $\mathbf{z}_v \in \mathcal{Z}$ and a multi-channel feature vector $\mathbf{x}_v \in \mathcal{X} \subset \mathbb{R}^{F \times C}$ and each edge $e \in \mathcal{E}$ is associated with a vector $\mathbf{w}_e \in \mathcal{W} \subset \mathbb{R}^W$. In regular graph learning, we learn a predictor $f$ to associate a node $\mathbf{x}_v, v \in \mathcal{V}'$ with a target $\mathbf{z}_v$, given graph $\mathcal{G}$, node features $\mathcal{X}$, edge vectors $\mathcal{W}$, and part of the targets $\mathbf{z}_v, v \in \mathcal{V} \setminus \mathcal{V}'$.

In lifelong graph learning, we have the same objective, but can only obtain the graph-structured data from a data continuum $\mathcal{G}_L = \{ (\mathbf{x}_i, t_i, \mathbf{z}_i, \mathcal{N}_{k=1:K}(\mathbf{x}_i), \mathcal{W}_{k=1:K}(\mathbf{x}_i))_{i=1:N} \}$, where each item is formed by a node feature $\mathbf{x}_i \in \mathcal{X}$, a task descriptor $t_i \in \mathcal{T}$, a target vector $\mathbf{z}_i \in \mathcal{Z}_{t_i}$, a $k$-hop neighbor set $\mathcal{N}_{k=1:K}(\mathbf{x}_i)$, and an edge vector set $\mathcal{W}_{k=1:K}(\mathbf{x}_i)$ associated with the $k$-hop neighbors. For simplicity, we will use the symbol $\mathcal{N}(\mathbf{x}_i)$ to denote the available neighbor set and their edges. We assume that every item $(\mathbf{x}_i, \mathcal{N}(\mathbf{x}_i), t_i, \mathbf{z}_i)$ satisfies $(\mathbf{x}_i, \mathcal{N}(\mathbf{x}_i), \mathbf{z}_i) \sim P_{t_i}(\mathcal{X}, \mathcal{N}(\mathcal{X}), \mathcal{Z})$, where $P_{t_i}$ is a probability distribution of a single learning task. In lifelong graph learning, we will observe, item by item, the continuum of the graph-structured data as

$$(\mathbf{x}_1, \mathcal{N}(\mathbf{x}_1), t_1, \mathbf{z}_1), \ldots, (\mathbf{x}_N, \mathcal{N}(\mathbf{x}_N), t_N, \mathbf{z}_N) \quad (1)$$

While observing (1), our goal is to learn a predictor $f_L$ to associate a test sample $(\mathbf{x}, \mathcal{N}(\mathbf{x}), t)$ with a target $\mathbf{z}$ such that $(\mathbf{x}, \mathcal{N}(\mathbf{x}), \mathbf{z}) \sim P_t$. Such test sample can belong to a task observed in the past, the current task, or a task observed (or not) in the future. The task descriptors $t_i$ is defined for compatibility with lifelong learning that requires them [26] but is not used in the experiments.

Note that samples are **not** drawn locally identically and independently distributed (i.i.d.) from a fixed probability distribution, since we don't know the task boundary. In the continuum, we only know the label of $\mathbf{x}_i$, but have no information about the labels of its neighbors $\mathcal{N}(\mathbf{x}_i)$. The items in (1) are unavailable once they are observed and

dropped. This is in contrast to the settings in [46], where all historical data are available during training. As shown in the experiments, lifelong graph learning in practice often requires that the number of GNN layers $L$ is larger than the availability of $K$-hop neighbors, *i.e.* $L > K$, which also leads many existing graph models inapplicable.

## 4. Feature Graph Network

To better show the relationship with a regular graph, we first review GCN [20]. Given an graph $\mathcal{G} = (\mathcal{V}, \mathcal{E})$ described in Section 3, the stacked node features can be written as $\mathbf{X} = [\mathbf{x}_1, \mathbf{x}_2, \cdots, \mathbf{x}_N]^{\mathrm{T}} \in \mathbb{R}^{N \times FC}$, where $\mathbf{x}_i \in \mathbb{R}^{FC}$ is a vectorized node feature. The GCN takes feature channel as $C = 1$, so that $\mathbf{x}_i \in \mathbb{R}^F, \mathbf{X} \in \mathbb{R}^{N \times F}$. The $l$-th graph convolutional layer is defined as

$$\mathbf{X}_{(l+1)} = \sigma \left( \hat{\mathbf{A}} \cdot \mathbf{X}_{(l)} \cdot \mathbf{W} \right), \quad (2)$$

where $\sigma(\,\cdot\,)$ is an activation function, $\mathbf{W} \in \mathbb{R}^{F_{(l)} \times F_{(l+1)}}$ is a learnable parameter, and $\hat{\mathbf{A}} \in \mathbb{R}^{N \times N}$ is a normalized adjacency for $\mathbf{A}$ (refer to [20] for details). A graph convolutional layer doesn't change the number of nodes (rows of $\mathbf{X}$) but change it feature dimension from $F_{(l)}$ to $F_{(l+1)}$.

GCN has been applied to many graph-structured tasks due to its simplicity and good generalization ability. However, the problems of GCN are also obvious. Besides the forgetting problem in lifelong learning, its node features in the next layer is a linear combination of the current layer, thus GCN and its variants cannot directly model the feature "interaction". To this end, we introduce the feature graphs by defining feature nodes and feature adjacency matrix.

### 4.1. Feature Nodes

Recall that each node $v$ in a regular graph $\mathcal{G} = (\mathcal{V}, \mathcal{E})$ is associated with a multi-channel feature vector $\mathbf{x} = \left[ \mathbf{x}_{[1,:]}^{\mathrm{T}}, \cdots, \mathbf{x}_{[F,:]}^{\mathrm{T}} \right] \in \mathbb{R}^{F \times C}$, where $\mathbf{x}_{[i,:]}$ is the $i$-th feature (row) of $\mathbf{x}$. An attribute feature graph takes the features of a regular graph as nodes. It can be defined as $\mathcal{G}^{\mathcal{F}} = (\mathcal{V}^{\mathcal{F}}, \mathcal{E}^{\mathcal{F}})$, where each node $v^{\mathcal{F}} \in \mathcal{V}^{\mathcal{F}}$ is associated with a feature $\mathbf{x}_{[i,:]}^{\mathrm{T}}$ and will be denoted as $\mathbf{x}_i^{\mathcal{F}}$. Intuitively, the number of nodes in the feature graph is the feature dimension in the regular graph, *i.e.* $|\mathcal{V}^{\mathcal{F}}| = F$, and the feature dimension in feature graph is the feature channel in a regular graph, *i.e.* $\mathbf{x}_i^{\mathcal{F}} \in \mathbb{R}^C$. Therefore, we define the feature nodes for feature graph as

$$\mathcal{V}^{\mathcal{F}} = \left\{ \mathbf{x}_1^{\mathcal{F}}, \mathbf{x}_2^{\mathcal{F}}, \cdots, \mathbf{x}_i^{\mathcal{F}}, \cdots, \mathbf{x}_F^{\mathcal{F}} \right\}. \quad (3)$$

In this way, for each node $v \in \mathcal{V}$, we have a feature graph $\mathcal{G}^{\mathcal{F}}$. We next establish their relationship by defining the feature adjacency matrices via feature cross-correlation.

### 4.2. Feature Adjacency Matrix

For each item in continuum (1), the edges between $\mathbf{x}$ and its neighbors $\mathcal{N}(\mathbf{x})$ imply the existence of correlations

Table 1. The relationship of a graph and feature graph.

| Graph | | Feature Graph | | Relationship | |
|---|---|---|---|---|---|
| Node | $\mapsto$ | Graph | Node classification | $\mapsto$ | Graph classification |
| Feature | $\mapsto$ | Node | Fixed Feature | $\mapsto$ | Fixed Nodes |
| Edge | $\mapsto$ | Edges | Growing Adjacency | $\mapsto$ | Multiple Adjacency |
| Graph | $\mapsto$ | Samples | Dynamic graph | $\mapsto$ | Multiple samples |

between their features. We model the feature adjacency as the correlation over the $k$-hop neighborhood $\mathcal{N}_k(\mathbf{x})$ and for each of the $c$ channels independently, where $c = 1, \ldots, C$:

$$\mathbf{A}_{k,c}^{\mathcal{F}}(\mathbf{x}) \triangleq \text{sgnroot}\left(\mathbb{E}_{\mathbf{y} \sim \mathcal{N}_k(\mathbf{x})}\left[w_{x,y}\mathbf{x}_{[:,c]}\mathbf{y}_{[:,c]}^{\text{T}}\right]\right), \quad (4)$$

where $w_{x,y} \in \mathbb{R}$ is the associated edge weight and $\text{sgnroot}(x) = \text{sign}(x)\sqrt{|x|}$ retains the magnitude of node features and the sign of their inner products. Note that $\mathbf{A}_{k,c}^{\mathcal{F}}$ preserves the connectivity information by only encoding information from connected nodes. For each sample $\mathbf{x}$, this produces $C$ matrices of size $F \times F$, where $F \ll N$ due to the lifelong learning settings. For undirected graphs, we change $\mathbf{x}_{[:,c]}\mathbf{y}_{[:,c]}^{\text{T}}$ to $(\mathbf{x}_{[:,c]}\mathbf{y}_{[:,c]}^{\text{T}} + \mathbf{y}_{[:,c]}\mathbf{x}_{[:,c]}^{\text{T}})$ for symmetry.

In practice, the expectation in (4) is approximated by averaging over the observed neighborhood:

$$\mathbb{E}\left[w_{x,y}\mathbf{x}_{[:,c]}\mathbf{y}_{[:,c]}^{\text{T}}\right] \approx \frac{\sum_{\mathbf{y} \in \mathcal{N}_k(\mathbf{x})} w_{x,y}\mathbf{x}_{[:,c]}\mathbf{y}_{[:,c]}^{\text{T}}}{|\mathcal{N}_k(\mathbf{x})|}. \quad (5)$$

In this way, the feature adjacency matrix is constructed dynamically and independently from neighborhood samples via (5), so that the continuum (1) is converted to graphs:

$$(\mathcal{G}_1^{\mathcal{F}}, t_1, \mathbf{z}_1), \ldots, (\mathcal{G}_i^{\mathcal{F}}, t_i, \mathbf{z}_i), \ldots, (\mathcal{G}_N^{\mathcal{F}}, t_N, \mathbf{z}_N) \quad (6)$$

where $\mathcal{G}_i^{\mathcal{F}} = \left(\mathcal{V}_i^{\mathcal{F}}, \mathbf{A}^{\mathcal{F}}(\mathbf{x}_i)\right)$. This means that our objective of learning a node predictor becomes learning a graph predictor $f^{\mathcal{F}} : \mathcal{G}^{\mathcal{F}} \times \mathcal{T} \mapsto \mathcal{Z}$ that predicts target $\mathbf{z}$ for test sample $(\mathcal{G}^{\mathcal{F}}, t)$ so that $(\mathcal{G}^{\mathcal{F}}, t, \mathbf{z}) \sim P_t^{\mathcal{F}}$. Note that the feature graphs in the new continuum (6) are still non-i.i.d.

In this way, a growing adjacency matrix is converted to multiple small adjacency matrices. Hence, a lifelong graph learning problem becomes a regular lifelong learning problem similar to [26] and the problem of increasing nodes can be solved by applying lifelong learning to the graph continuum (6). To be more clear, we list their detailed relationship in Table 1, where the arrow $\mapsto$ refers to the conversion from a regular graph to multiple feature graphs.

## 4.3. Feature Graph Layers

Since feature graph is a new topology given by feature adjacency matrix, we are able to define many different types of layers. In this section we present three types of layers.

### 4.3.1 Feature Broadcast Layer

Inspired by GCN, the $l$-th broadcast layer is defined as

$$\mathbf{x}_{(l+1)}^{\mathcal{F}} = \sigma\left(\hat{\mathbf{A}}_k^{\mathcal{F}} \cdot \mathbf{x}_{(l)}^{\mathcal{F}} \cdot \mathbf{W}^{\mathcal{F}}\right), \quad (7)$$

where $\sigma(\cdot)$ is a non-linear activation function, $\hat{\mathbf{A}}_k^{\mathcal{F}}$ is the associated normalized feature adjacency matrix, and $\mathbf{W} \in \mathbb{R}^{C_{(l)} \times C_{(l+1)}}$ is a learnable parameter. For simplicity, the channel $c$ is left out in (7) and the layer channel is broadcasted independently. It is worth noting that, although the definition of (7) appears similar to (2), they have different dimension and represent different meanings.

### 4.3.2 Feature Transform Layer

Similar to graph convolutional layer (2), the feature broadcast layer doesn't change the number of feature nodes. However, this is not always necessary, as the objective has been turned into graph classification. Therefore, we define a feature transform layer which can change the number of feature nodes and will be helpful for further reducing the number of learnable parameters. Different from the feature broadcast layer, we need to re-calculate the feature adjacency matrices from transformed neighbors. Therefore, given the feature graph $\mathcal{G}^{\mathcal{F}}$, the $l$-th feature transform layer can be defined as

$$\mathbf{A}_{(l)}^{\mathcal{F}}(\mathbf{x}) \triangleq \mathbf{A}_k^{\mathcal{F}}\left(\mathbf{x}_{(l)}, \mathbf{y}_{(l)}\right), \quad \forall \mathbf{y} \in \mathcal{N}_k(\mathbf{x}), \quad (8a)$$

$$\mathbf{x}_{(l+1)}^{\mathcal{F}} = \sigma\left(\mathbf{W}^{\mathcal{F}} \cdot \hat{\mathbf{A}}_{(l)}^{\mathcal{F}}(\mathbf{x}) \cdot \mathbf{x}_{(l)}^{\mathcal{F}}\right), \quad (8b)$$

$$\mathbf{y}_{(l+1)}^{\mathcal{F}} = \sigma\left(\mathbf{W}^{\mathcal{F}} \cdot \hat{\mathbf{A}}_{(l)}^{\mathcal{F}}(\mathbf{x}) \cdot \mathbf{y}_{(l)}^{\mathcal{F}}\right), \quad (8c)$$

where $\mathbf{W}^{\mathcal{F}} \in \mathbb{R}^{F_{(l+1)} \times F_{(l)}}$ is a learnable parameter, $\sigma(\cdot)$ is a non-linear activation function, and $\hat{\mathbf{A}}_{(l)}^{\mathcal{F}}(\mathbf{x}) \in \mathbb{R}^{F_{(l)} \times F_{(l)}}$ is the normalized feature adjacency $\mathbf{A}_{(l)}^{\mathcal{F}}(\mathbf{x})$. The node features sometimes can be smoothed due to graph propagation, hence we can replace $\hat{\mathbf{A}}_{(l)}^{\mathcal{F}}(\mathbf{x}) \cdot \mathbf{y}_{(l)}^{\mathcal{F}}$ in (8c) to $[\hat{\mathbf{A}}_{(l)}^{\mathcal{F}}(\mathbf{x}) \cdot \mathbf{y}_{(l)}^{\mathcal{F}}, \mathbf{y}_{(l)}^{\mathcal{F}}]$ by concatenating input features to prevent over-smoothing.

### 4.3.3 Feature Attention Layer

In the cases that a graph is fully connected or the edges have no weights, $w_{x,y}$ in (4) will be not well defined. Prior methods often rely on an attention mechanism, *e.g.* GAT [42], to focus on important neighbors. Inspired by this, we define the edge weights as an attention in (9).

$$w_{x,y} = \frac{\exp(e_{x,y})}{\sum_{z \in \mathcal{N}(z)} \exp(e_{x,z})}, \quad (9a)$$

$$e_{x,y} = \text{LeakyReLU}(\mathbf{a}_x^T \mathbf{x} + \mathbf{a}_y^T \mathbf{y} + b), \quad (9b)$$

where $\mathbf{a}_x, \mathbf{a}_y \in \mathbb{R}^F, b \in \mathbb{R}$ are learnable attention parameters. We can also construct other types of layers based on the topology $\mathcal{G}^{\mathcal{F}}$, *e.g.* extending convolution to kervolution [44] to improve the model expressivity or combining the pagerank algorithm [22] to further reduce feature over-smoothness. In this paper, we will mainly demonstrate the effectiveness of the three types of layers introduced above.

## 4.4. Analysis and Computational Complexity

We next provide an intuitive explanation for feature graphs. A feature graph doesn't retain the physical meaning of its regular graph. Take a social network as an example, in a regular graph, each user is a node and user connections are edges. In feature graphs, each user *is* a graph, while the user features, including the users' age, gender, *etc.*, are nodes. Therefore, user behavior prediction becomes graph classification based on node information and new users simply become multiple training samples. Since the number of user features is more stable than the number of users, the size of a feature graph is more stable than its regular graph. This simplifies the learning on growing graphs dramatically by reducing the problem to the sample-incremental learning.

On the other hand, a regular graph usually assumes that some useful information of a node is often encoded into its neighbors, thus graph propagation is able to improve the model performance. A feature graph has the same assumption. However, feature graph doesn't directly propagate the neighbor features, but encode the neighbors into the feature adjacency matrices (4). Regardless of the nonlinear function, existing methods propagate neighbor features as $\hat{\mathbf{A}}\mathbf{X}$, where $\mathbf{X} = [\mathbf{x}_1, \cdots, \mathbf{x}_n]$, meaning they can only propagate information element-wisely, as features in the next layer are weighted average of the current features [20, 22, 42, 58].

In contrast, each feature graph propagate features via $\hat{\mathbf{A}}^{\mathcal{F}}\mathbf{x}^{\mathcal{F}}$, which explicitly model "interaction" between features. In this sense, feature graph doesn't lose information of edges but encode them into the feature adjacency. This explains its superiority over regular graph models in some cases of conventional graph learning as shown in Section 5. Take a citation graph as example, a keyword in an article may influence another keyword in the article citing the former. However, element-wise graph propagation like (2) cannot explicitly model this relationship [20, 22, 42, 58]. Although feature graph is also suitable for regular graph learning, we mainly focus our discussion on lifelong graph learning.

Feature graphs have a low computational complexity. Concretely, the complexity for calculating the feature adjacency is $\mathcal{O}(nF_{(l)}^2)$, where $n$ is the expected number of node neighbors in the continuum. Therefore, regardless of the number of layers which is a constant for specific models, the complexity of graph propagation for a feature broadcast layer and a feature transform layer is of $\mathcal{O}(F_{(l)}^2 C_{(l)} C_{(l+1)})$ and $\mathcal{O}(F_{(l)} F_{(l+1)} C_{(l)}^2)$, respectively. If we take $E_{(l)} = F_{(l)} C_{(l)}$ as the number of elements of the features, the complexity for calculating each sample is roughly of $\mathcal{O}(E^2 + nF^2)$, where we leave out the layer index for simplicity. Note that they do not depend on the task number, thus feature graphs have constant complexity with the increased learning tasks.

FGN is applicable even when the number of features $F$ is very large. In this case, we often use a feature transformation layer (8) as a feature extractor to project the raw features onto a lower dimension. For example, we reduced the feature dimension on Cora from $F_{(1)} = 1433$ to $F_{(2)} = 10$, which is also adopted by GCN, GAT, APPNP, SAGE, etc.

## 5. Experiments

**Implementation Details** We perform comprehensive tests on popular graph datasets including citation graph Cora, Citeseer, Pubmed [38], and ogbn-arXiv [29]. For each dataset, we construct two different continuum: data-incremental and class-incremental. In data-incremental tasks, all samples are streamed randomly, while in class-incremental tasks, all samples from one class are streamed before switching to the next class. In both tasks, each node can only be present to the model once, *i.e.*, a sample cannot be used to update the model parameters again once dropped. For this experiment, we implement a two-layer feature graph network in PyTorch [31] and adopt the SGD [19] optimizer. See further details in Appendix A. We choose the most popular graph models including GCN [20], GAT [42], SAGE [13], and APPNP [22] as our baseline. Comparison with other methods such as SAINT [12] are omitted as they require a pre-processing of the entire dataset, which is incompatible with the lifelong learning setting. The overall test accuracy after learning all tasks is adopted as the evaluation metric [2].

**Sequence Invariant Sampling** We adopt the lifelong learning method described in [2] with a minor improvement for all the baseline models. As reported in the Section 4.2 of [2], it tends to select less earlier items in the continuum, which discourages the memorization of earlier knowledge. We find that this is because a uniform sampling is adopted for all items. To compensate for such effect, we set a customized selection probability for different items (See Appendix B).

**Performance** Lifelong learning has a performance upper bound given by the regular learning settings, thus the performance in regular learning is also an important indicator for the effectiveness of feature graph. We present the overall performance of regular learning in Table 2 and denote this task as "R", where the same dataset settings are adopted for different models. To demonstrate the necessity of graph models, we also report the performance of MLP, which neglects the graph edges. It can be seen that feature graph achieves a comparable overall accuracy in all the datasets. This means that feature graph may also be useful for regular graph learning. We next show that feature graph in lifelong learning is approaching this upper bound and dramatically alleviate the issue of "catastrophic forgetting".

The overall averaged performance on the data-incremental tasks is reported as task of "D" in Table 2, where all performance is an average of three runs. We use a memory size of 500 for the datasets Cora, Citeseer and Pubmed, and a memory size of 512 for ogbn-arXiv. It is also worth noting that we have a low standard deviation, which means

Table 2. Overall performance comparison on all the datasets in Section 5.[†]

| Method | Task[‡] | Cora | Citeseer | Pubmed | ogbn-arXiv |
|---|---|---|---|---|---|
| MLP | | 0.673 | 0.702 | 0.832 | 0.462 |
| GCN | | 0.850 | 0.768 | 0.868 | 0.557 |
| SAGE | R | 0.850 | 0.768 | 0.858 | 0.615 |
| GAT | | 0.877 | 0.776 | 0.872 | 0.619 |
| APPNP | | 0.883 | 0.777 | 0.870 | 0.615 |
| **FGN** | | **0.887** | **0.785** | **0.884** | **0.631** |
| MLP | | $0.652 \pm 0.006$ | $0.707 \pm 0.008$ | $0.812 \pm 0.006$ | $0.349 \pm 0.006$ |
| GCN | | $0.827 \pm 0.003$ | $0.760 \pm 0.004$ | $0.845 \pm 0.002$ | $0.397 \pm 0.012$ |
| SAGE | D | $0.778 \pm 0.016$ | $0.712 \pm 0.015$ | $0.816 \pm 0.015$ | $0.531 \pm 0.006$ |
| GAT | | $0.857 \pm 0.007$ | $0.735 \pm 0.009$ | $0.847 \pm 0.002$ | $0.512 \pm 0.003$ |
| APPNP | | $0.861 \pm 0.001$ | $0.725 \pm 0.016$ | $0.851 \pm 0.003$ | $0.511 \pm 0.002$ |
| **FGN** | | **$0.870 \pm 0.011$** | **$0.762 \pm 0.003$** | **$0.872 \pm 0.009$** | **$0.555 \pm 0.010$** |
| MLP | | $0.558 \pm 0.010$ | $0.597 \pm 0.012$ | $0.816 \pm 0.013$ | $0.237 \pm 0.011$ |
| GCN | | $0.796 \pm 0.022$ | $0.695 \pm 0.003$ | $0.851 \pm 0.012$ | $0.328 \pm 0.011$ |
| SAGE | C | $0.826 \pm 0.014$ | $0.716 \pm 0.001$ | $0.841 \pm 0.011$ | $0.455 \pm 0.003$ |
| GAT | | $0.833 \pm 0.032$ | $0.698 \pm 0.009$ | $0.833 \pm 0.006$ | $0.409 \pm 0.006$ |
| APPNP | | $0.818 \pm 0.017$ | $0.689 \pm 0.003$ | $0.844 \pm 0.013$ | $0.423 \pm 0.007$ |
| **FGN** | | **$0.853 \pm 0.014$** | **$0.733 \pm 0.009$** | **$0.857 \pm 0.003$** | **$0.494 \pm 0.003$** |

[†]We denote the best performance in **bold** and second best with underline for each task. Due to the settings of lifelong learning, dataset split is different from the original one, as we only test the final optimized model and don't need a validation set (Appendix A).
[‡]Task of "R", "D", "C" denote regular, data-incremental, and class-incremental learning, respectively.

Table 3. Average class forgetting rate in task "C" (%).

| Model | Cora | Citeseer | Pubmed | ogbn-arXiv |
|---|---|---|---|---|
| APPNP | 6.38 | 8.03 | 3.61 | 16.3 |
| GAT | 5.50 | 7.05 | 3.50 | 17.7 |
| **FGN** | **1.35** | **0.41** | **3.49** | **14.8** |

Table 4. The number of model parameters used in Section 5.

| Model | MLP | GCN | SAGE | GAT | APPNP | FGN |
|---|---|---|---|---|---|---|
| # Parameters | 37,888 | 37,888 | 37,932 | 38,700 | 38,184 | **21,872** |

that its performance is stable and not sensitive to the model initialization. Samples in the class-incremental tasks are non-i.i.d. and we have no prior knowledge on the task boundary, thus it is generally *more difficult* than data-incremental tasks. The overall test accuracy on class-incremental tasks are listed as "C" in Table 2. Similar to the data-incremental tasks, we use a memory size of 500 for Cora, Pubmed, and Citeseer, and a memory size of 4096 for ogbn-arXix, which is grouped into 8 tasks, each of which contains 5 classes. Despite the difficulty, FGN still obtains higher performance on ogbn-arXix in task "C", compared to other state-of-the-art methods, which verifies its superiority in lifelong graph learning. We further report the forgetting rate [6] in the task "C" for the best models in Table 3, which is the average precision drop compared to the regular learning. It can be seen that FGN achieves the least forgetting rate, which demonstrates its effectiveness in lifelong graph learning.

**Memory Efficiency** We report the number of learnable parameters in Table 4. Due to its simplicity, FGN only requires 42% less parameters compared to GCN and SAGE, which demonstrates its memory efficiency.

## 6. Distributed Human Action Recognition

**Implementation Details** To demonstrate the flexibility of FGN, we apply it to an application, *i.e.* distributed human action recognition using wearable motion sensor networks in Figure 2a (Data from [55]). Five sensors, each of which consists of a triaxial accelerometer and a biaxial gyroscope, are located at the left and right forearms, waist, left and right ankles, respectively. Each sensor produces 5 data streams and totally $5 \times 5$ data streams is available. The stream is recorded at 30Hz and is comprised of human subjects with ages from 19 to 75 and 13 daily action categories, including rest at standing (ReSt), rest at sitting (ReSi), rest at lying (ReLi), walk forward (WaFo), walk forward left-circle (WaLe), walk forward right-circle (WaRi), turn left (TuLe), turn right (TuRi), go upstairs (Up), go downstairs (Down), jog (Jog), jump (Jump), and push wheelchair (Push).

We take every 25 sequential data points from a single sensor as a node and perform recognition for every 50 sequential points (1.67s) shown in Figure 2b, which is also adopted by [45]. This results in a temporally growing graph: Each node is associated with a multi-channel ($C = 5$) 1-D signal $\mathbf{x}_t^i \in \mathbb{R}^{25 \times 5}$, where $i = 1, 2 \ldots 5$ is the sensor index

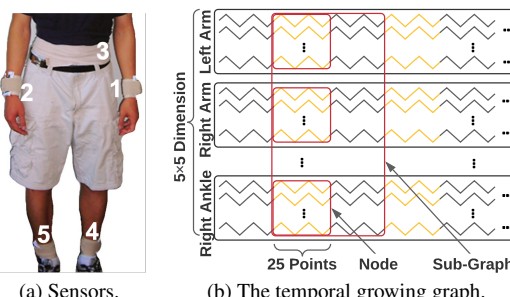
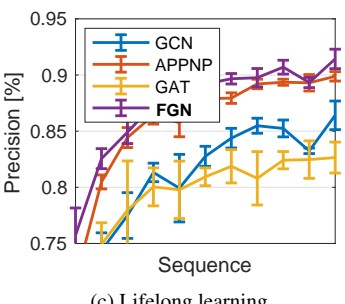
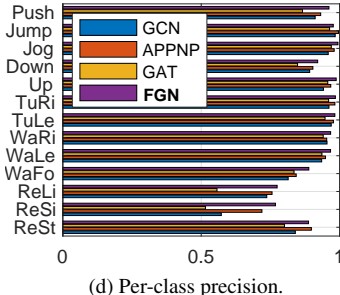

(a) Sensors.    (b) The temporal growing graph.    (c) Lifelong learning.    (d) Per-class precision.

Figure 2. The wearable sensor networks, the streaming graph, precesion during learning, and the final performance comparison.

Table 5. Backward maximum forgetting rate of all models on all actions (%).

| Model | ReSt | ReSi | ReLi | WaFo | WaLe | WaRi | TuLe | TuRi | Up | Down | Jog | Jump | Push | **Overall** |
|---|---|---|---|---|---|---|---|---|---|---|---|---|---|---|
| GCN | 5.34 | 12.49 | 0.99 | 1.32 | 2.88 | 0.59 | 0.89 | 1.58 | 1.30 | 2.81 | 1.24 | 0.26 | 5.31 | 2.85 |
| APPNP | 0.81 | 2.57 | 7.17 | 1.37 | 2.70 | 0.80 | 0.94 | 0.44 | 0.11 | 3.97 | 0.99 | 0.00 | 3.66 | 1.96 |
| GAT | 1.34 | 6.12 | 4.23 | 0.00 | 2.79 | 0.76 | 0.82 | 0.83 | 0.81 | 2.27 | 0.62 | 0.87 | 4.79 | 2.02 |
| **FGN** | 0.19 | 0.56 | 3.32 | 0.95 | 2.67 | 0.53 | 1.15 | 1.04 | 0.00 | 3.52 | 0.46 | 1.02 | 2.28 | **1.36** |

Table 6. Precision comparison on the action recognition.

| | MLP | GCN | APPNP | GAT | **FGN** |
|---|---|---|---|---|---|
| Regular | 0.645±0.046 | 0.881±0.016 | 0.942±0.006 | 0.911±0.004 | **0.954±0.004** |
| Lifelong | -† | 0.864±0.013 | 0.898±0.004 | 0.826±0.014 | **0.914±0.009** |

†The notation "-" means that we cannot obtain meaningful performance on this task.

and $t = 1, 2 \ldots T$ is the time index. We assume all nodes at adjacent time index are connected, *i.e.* for all $t$, nodes $(\mathbf{x}_t^1, \ldots, \mathbf{x}_t^5, \mathbf{x}_{t+1}^1, \ldots, \mathbf{x}_{t+1}^5)$ form a fully connected sub-graph. Therefore, the problem of human action recognition becomes a problem of sub-graph (10 nodes) classification.

**Performance** We list the overall performance of regular and lifelong learning in Table 6, which is obtained from an average of three runs. It can be seen that FGN achieves the best overall performance in regular learning and a much higher performance in lifelong learning compared to the state-of-the-art methods. This means that FGN has a lower forgetting rate, demonstrating its effectiveness.

We present the overall test accuracy during the lifelong learning process in Figure 2c. It can be seen that FGN has a much higher and stable performance than all the other methods. We also shown their final per-class precision in Figure 2d, which indicates that FGN achieves a much higher performance in nearly all the categories. Note that all models have a relatively low performance on ReSt, ReSi, and ReLi. This is because their signals are relatively flat and similar due to their static posture. This phenomenon is also observed by a template matching method KCC [45]. Note that we don't directly compare against to KCC, since KCC mainly performs matching for single subject, while we aim for prediction across different subjects.

We argue that the superior performance of FGN is because of that FGN is able to explicitly model the relationship

of different features. Take the walking action as an example, it is well known that the movement of a left arm is always related to the movement of the right leg. Such information can be explicitly modeled by the cross-correlation in the feature adjacency matrix. Moreover, it is also able to model the feature relationship at different time step, *e.g.* the movement of a left arm at time $t$ is also related to the movement of the right arm at time $t + 1$. As aforementioned, FGN can explicitly learn such relationship with $k = 2$ in (4), while the other methods can't directly do this. Besides, FGN also achieves the least forgetting rate in Table 5 over all classes, which is the difference between the final and best performance during the entire learning process.

## 7. Image Feature Matching

**Implementation Details** We next extend FGN to a more challenging application, *i.e.* image feature matching. It is crucial for many 3-D computer vision tasks including simultaneous localization and mapping (SLAM). As shown in Figure 4, the interest point and their descriptors form an infinite temporal growing graph, in which the feature points are nodes and their descriptors are the node features as defined in Section 3. In this way, the problem of feature matching becomes edge prediction for a temporal growing graph. We next show that the performance of SuperGlue [37], which used a regular graph attention model for feature matching, can be simply improved by changing the graph-attention matcher to our proposed FGN. For simplicity, we adopt a framework similar to that of SuperGlue but removed the cross-attention layers in the matching network. In image feature matching, we normally have a layered temporal graph, where edge weights are undefined. Therefore, we construct

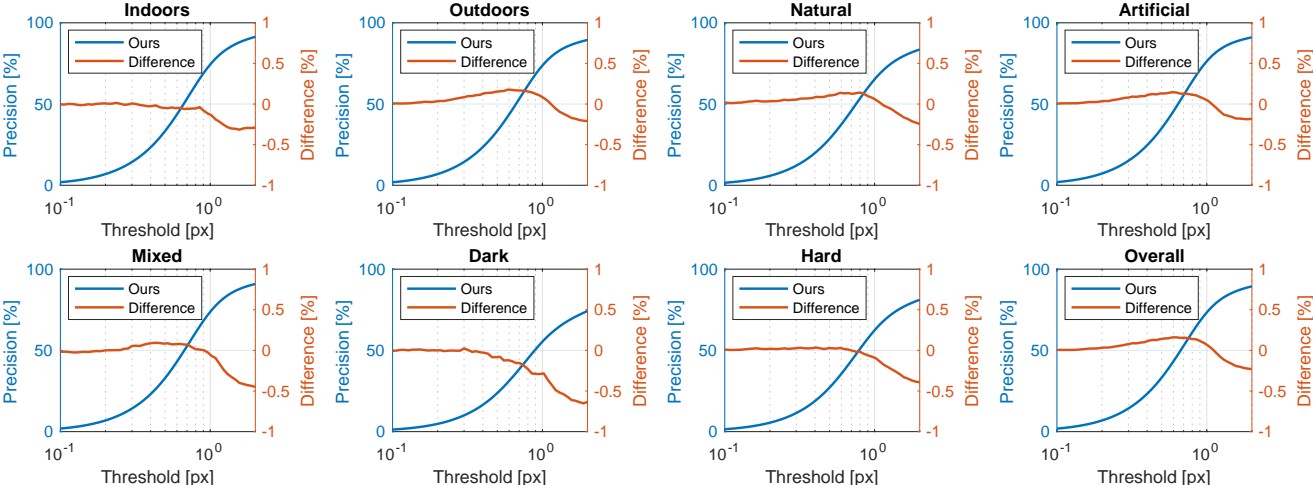

Figure 3. The matching precision of our method (blue) and performance gap between our method and GAT (orange) at different level of tolerances. Our method outperforms GAT at sub-pixel precision in mildly difficult categories.

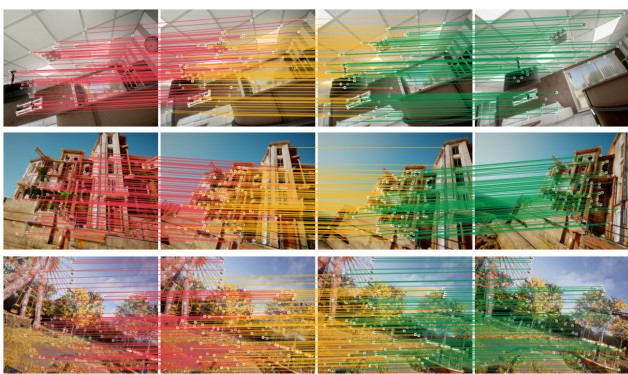

Figure 4. Matching examples on TartanAir dataset. Feature matching is a problem of edge prediction for temporal growing graph.

FGN by concatenating two feature broadcast layers (7) with the attention edge weights $w_{x,y}$ defined in (9).

**Performance** We categorize the 80 test sequences from TartanAir dataset [47] into groups based on their characteristics. Figure 4 shows several examples of consecutive matching in Indoors, Outdoors&Artificial, and Outdoors&Mixed environments. We report the mean matching error in Table 7, compared with the GAT-based matcher. It can be observed that our method outperforms GAT by a noticeable margin on all categories, especially on the difficult ones (Outdoors, Natural, Hard). Specifically, FGN achieves an overall error reduction of 24.2% compared with GAT.

We further report the difference in error landscape of FGN and GAT in Figure 3. Although there is no significant difference in matching precision at the single-pixel level, our method experiences a greater boost in precision in the sub-pixel region when error tolerance increases. This effect is especially noticeable for the Artificial and Outdoors categories. This suggests that our method is more advantageous at identifying high quality matches in mildly difficult envi-

Table 7. Mean Error (pixels) in image feature matching.†

| Category | I | O | N | A | M | D | H | Overall |
|---|---|---|---|---|---|---|---|---|
| # Sequence | 17 | 63 | 19 | 53 | 11 | 6 | 27 | 80 |
| GAT | 1.70 | 2.73 | 4.17 | 2.31 | 2.19 | 6.00 | 4.68 | 2.64 |
| FGN (Ours) | **1.52** | **2.05** | **2.88** | **1.85** | **1.56** | **4.75** | **3.25** | **2.00** |
| Reduction (%) | -10.6 | -24.9 | -30.9 | -19.9 | -28.8 | -20.8 | -30.6 | -24.2 |

†Scene categories include: (I)ndoor and (O)utdoor, (N)aturalistic (woods and sea floor), (A)rtificial (streets and buildings), (M)ixed (containing both natural and artificial objects), (D)ark (poor lighting condition), (H)ard (violent motion and/or complex textures).

ronments, which is beneficial for downstream tasks such as simultaneous localization and mapping (SLAM).

## 8. Conclusion

In this paper, we focus on the problem of lifelong graph learning and propose the feature graph as a new graph topology, which solves the challenge of increasing nodes in a streaming graph. It takes features of a regular graph as nodes and takes the nodes as independent graphs. To construct the feature adjacency matrices, we accumulate the cross-correlation matrices of the connected feature vectors to model feature interaction. This successfully converts the original node classification to graph classification and turns the problem from lifelong graph learning into regular lifelong learning. The comprehensive experiments show that feature graph achieves superior performance in both data-incremental and class-incremental tasks. The applications on action recognition and feature matching demonstrates its superiority in lifelong graph learning. To the best of our knowledge, feature graph is the first work to bridge graph learning to lifelong learning via a novel graph topology.

# Acknowledgment

This work was sponsored by ONR grant #N0014-19-1-2266 and ARL DCIST CRA award W911NF-17-2-0181.

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

## A. Dataset Details

We perform comprehensive experiments on popular graph datasets including citation graph Cora, Citeseer, Pubmed, and ogbn-arXiv [28, 29, 38]. Their statistics are listed in Table 8. Note that the dataset split is a little different from their original settings, as we can only test the final optimized model due to the requirement of lifelong learning, thus we don't need a validation set. For Cora, Citeseer, and Pubmed, the model consists of two feature broadcast layers (7) with $C_{(1)} = 1$ and $C_{(2)} = 2$ channels each. For obgn-arXiv, we found that node features can be easily smoothed due to multiple feature propagation, hence we use the feature transform layers to concatenate its input features. We take the one-hot vector as the target vector $\mathbf{z}_i$, adopt the cross-entropy loss, and use the softsign [11] function $\sigma(x) = x/(1+|x|)$ as the non-linear activation.

Table 8. The statistics of the datasets used for lifelong learning.

| Dataset | Nodes | Edges | Classes | Features | Labels |
|---|---|---|---|---|---|
| Cora | 2,708 | 5,429 | 7 | 1,433 | 0.421 |
| Citeseer | 3,327 | 4,732 | 6 | 3,703 | 0.337 |
| Pubmed | 19,717 | 44,338 | 3 | 500 | 0.054 |
| obgn-arXiv | 169,343 | 1,166,243 | 40 | 128 | 0.5 |

## B. Proof of Sequence Invariant Sampling

Let $P$ be the probability that the observed items are selected at time $t$, thus the probability that one item is still kept in the memory after $k$ selection is $P^k$. This explains that earlier items have lower probability to be kept in the memory and this phenomenon was reported in the Section 4.2 of [2]. To compensate for such effect and

**Proposition B.1.** *To ensure that all items in the continuum have the same probability to be kept in the memory at any time $t$, we can set the probability that the $n$-th item is selected*

at time $t$ as

$$P_n(t) = \begin{cases} 1 & t \leqslant M \\ M/n & t > M,\ t = n \\ (n-1)/n & t > M,\ t > n \end{cases}, \quad (10)$$

where $M$ denotes the memory size.

*Proof.* It is obvious for $t \leqslant M$ as we only need to keep all items in the continuum. For $t > M$, the probability that the $n$-th item is still kept in the memory at time $t$ is

$$\begin{aligned} P_{n,t} &= P_n(n) \cdot P_n(n+1) \cdots P_n(t-1) \cdot P_n(t), \\ &= \frac{M}{n} \cdot \frac{n}{n+1} \cdot \cdots \cdot \frac{t-2}{t-1} \cdot \frac{t-1}{t}, \\ &= \frac{M}{t}. \end{aligned} \quad (11)$$

This means the probability $P_{n,t}$ is irrelevant to $n$ and all items in the continuum share the same probability. In practice, we always keep $M$ items and sample balanced items accross classes. $\square$

## C. Distributed Human Action Recognition

**Implementation** In practice, the temporal growing graph can only be learned sequentially, thus we take the first 80% of each sequence for training and the remaining 20% for testing. Specifically, we define the radius of neighborhood as the temporal distance. Therefore, all the nodes at the same instant are 1-hop neighbors of each other. For each feature graph we have $K = 2$ in the continuum (1). We construct FGN using two feature transform layers (8) with attention weights (9) and one fully connected layer to predict for the sub-graph classification. For fairness, we use $C_{(1)} = 5$, $F_{(1)} = 25$, $C_{(2)} = 32$, and $F_{(2)} = 12$ for all models. In the experiments, we find that GCN, APPNP obtain the best overall performance using the SGD optimizer, while MLP, GAT, and FGN performed the best using the Adam optimizer. **Running time** We also report the average running time for the models in Table 9. Note that the efficiency of FGN is on par with other methods. Considering that FGN has a much better performance, we believe that FGN is more promising.

Table 9. Running time comparison on the action recognition.

| | MLP | GCN | APPNP | GAT | **FGN** |
|---|---|---|---|---|---|
| Runtime (ms) | 3.51 | 5.38 | 5.36 | 5.48 | 5.76 |

## D. Image Feature Matching

Although many hand-crafted feature descriptors such as SIFT [27] and ORB [34] have been proposed decades ago, their performance is still unsatisfied for large view

point changes. Due to the well generalization ability, deep learning-based feature detectors have received increasing attentions. For example, SuperPoint [9] introduced a self-supervised framework for extracting interest point detectors and descriptors. SuperGlue [37] introduced graph attention model into SuperPoint for feature matching.

**Implementation** In the experiments, we adopt $C = 1$, $K = 1$, $F = 256$ in both FGN and FGN for fairness. The training loss function is adapted from [30] which maximizes the likelihood of predicting similar node embeddings corresponding to their spatial location. We recommend the readers refer to SuperPoint [9], SuperGlue [37], and [30] for more details of the loss functions.

**Dataset** We perform training and evaluation on the TartanAir dataset [47]. TartanAir is a large (about 3TB) and very challenging visual SLAM dataset consisting of binocular RGB-D video sequences together with additional per-frame information such as camera poses, optical flow, and semantic annotations. The sequences are rendered in AirSim [39], a photo-realistic simulator, which features modeled environments with various themes including urban, rural, nature, domestic, public, sci-fi, *etc*. Figure 4 contains several example video frames from TartanAir. The dataset is collected such that it covers challenging viewpoints and diverse motion patterns. In addition, the dataset also includes other traditionally challenging factors in SLAM tasks such as moving objects, changing lighting conditions, and extreme weather. We randomly select 80% of the sequences for training and take the remaining for testing. We recommend the readers refer to [47] for more details of the dataset.

## E. Limitation

Although we have shown that FGN outperformed the SOTA methods in node classification, sub-graph classification, and edge prediction, it also has several limitations. **First**, our current implementation is not vectorized for taking a varying number of neighbors, which is less computationally efficient. **Second**, in the experiments, we assumed scalar edge weights, while in the general case, the graph edge weights are represented by a vector, as defined in the feature graphs. **Third**, since our main contribution is the novel graph topology, *i.e.*, feature graph, we mainly compared it with the SOTA graph models such as GAT by applying an off-the-shelf lifelong learning algorithm. However, it might also be applicable to other lifelong learning algorithms. In the future, we plan to fully optimize the codes, extend it to applications with vector edge weights, and apply more lifelong learning algorithms.

