# OpenReview forum: "Bridging Graph Network to Lifelong Learning with Feature Interaction"
_ICLR.cc/2021/Conference — Reject_

### Official Review · AnonReviewer4 · 2020-10-25
**The idea of this work is novel, but the paper is not clear enough.**

**Rating:** 4
**Confidence:** 4

**Review:**

This paper aims to extend GCN to the lifelong learning setting. The idea is to transform the nodes of a graph into feature graphs where each feature is a node and the edges represent feature correlations estimated from the K-hop neighborhood of a node in the original graph.

[Pros] The idea of converting nodes of a graph into feature graphs is novel and interesting (as far as I know). In this way, the model depends on feature graphs rather than the original graph which continually grows in the online learning setting. This breaks down the original graph into individual nodes and their contexts which are then fed to the model to generate results. Very interesting.

[Cons] However, some major issues still hinder the paper from publication:
(1) There is a lack of necessary details. First, the constructed adjacency matrix $A^F_{k,c}$ (Eq.(5)) has two subscripts $k$ and $c$. But the subscript $c$ just disappears in Eq.(9) without any explanation. Therefore, it is not known how the final adjacency matrix is built in Eq.(9) from Eq.(5). Second, how do you set the edge wights of the original graph? Moreover, at first the problem formulation says each edge is associated with a weight vector, but it seems finally it is just a scalar. This is a little confusing. The above issues affect the reproducibility of this work.
(2) The notations are hard to follow. For example, in the definition of the graph lifelong learning, the weight vector set $\mathcal{W}$ is said to contain weight vectors associated with the $K$-hop (Strictly speaking, I think here it should be $K$ rather than $k$. More about this issue below) neighbors. Does this mean these weights are associated with nodes rather than edges? According to the notation of $\mathcal{W}$ ($k=1:K$), I think $\mathcal{W}$ contains neighbors up to $K$-hop rather than only the $K$-th hop. The use of $k$ and $K$ is very confusing, which makes related content hard to understand. Another question is, how do you set the parameter $K$? The last sentence said “a maximum $k=1$ is required...”. I am not sure whether this is how $K$ is set. Furthermore, what do you mean by maximum? Is it possible that $K=0$? The last question is, how do you set the target vector $z$ for the training data? I cannot find the related content.
(3) What is the motivation of the design of the feature transform layer? Why do we need to change the number of feature nodes? The motivation is not clear. Furthermore, there is no comparison to using broadcast layers in experiments. In the appendix F, why do you use the Flickr dataset for evaluating transform layers? Why not use this dataset in the main experiments?

Some minor issues:
(1) The writing needs some improvement. Some language errors include “an importance”, two sentences in a sentence.
(2) What is the motivation of using sgnroot in Eq.(5)?
(3) Traditional methods also exploit the interactions between features in a neighborhood, by concatenating feature vector of the target node and the aggregated neighborhood feature vector and do affine transformation. Hence, I think “existing methods cannot model feature interactions well” is not precise. Besides, “useful information might be encoded in one’s neighbor features” is vague. Some concrete examples are needed to motivate the modeling of feature interactions in node neighborhoods.

---

> ### Author Response · Authors · 2020-11-17
> **To Reviewer 4**
>
> We are excited that R4 believes that the idea of converting nodes of a graph into feature graphs is novel and very interesting. We next address your questions, respectively.
>
> ---
> Q1: The constructed adjacency matrix $A_{k,c}^F$  in (5) has subscripts  $k$ and $c$. But $c$ disappears in Eq. (9) without explanation. Therefore, it is not known how the final adjacency matrix is built in (9) from (5).
> - In (5), $c$ is the channel index. Equation (9) is the definition of feature broadcast layer for **a single channel**, thus $c$ is not included in (9)  for simplicity. The layer computation is conducted in separated channels then the outputs are concatenated to produce the signal output.
> - The graph data in experiment contains only 1 channel. However, we still give the multi-channel definition in (5) for compatibility with multi-channel signals, e.g., images have 3 channels, which will be useful for future works.
> - Thanks for your suggestions, we have updated Sec 4.3 to make this more clear.
>
> ---
> Q2: How do you set the edge weights of the original graph? Moreover, at first the problem formulation says each edge is associated with a weight vector, but it seems finally it is just a scalar. This is a little confusing. The above issues affect the reproducibility of this work.
> - The citation graph used in experiments have a predefined edge weight: 1 or 0 indicates whether two articles (nodes) are references of each other. The flickr dataset also has a predefined weight: 1 or 0 indicate whether two images (nodes) are from the same location, submitted to the same gallery, group, or set, etc.
> - We define the multi-channel edge weights in (5) also for compatibility with such graph data, but not used in this paper. In this way, we expect our method can inspire more research in the future.
> - We have updated the paper to clarify. We will release all source codes and pre-trained models to ensure the reproducibility of this work.
>
> ---
> Q3: In the definition of the graph lifelong learning, the weight vector set $\mathcal{W}$ is said to contain weight vectors associated with the K-hop neighbors. Does this mean these weights are associated with nodes rather than edges?
> - The weights are still associated with edges, while its availability in the continuum in Eq. (1) is associated with the nodes, as defined in the continuum.
>
> ---
> Q4: According to the notation of W (k=1:K), I think  W contains neighbors up to  K-hop rather than only the  K-th hop. The use of k and K is very confusing, which makes related content hard to understand. How do you set the parameter K? The last sentence said “a maximum k=1 is required...”. I am not sure whether this is how K is set. Furthermore, what do you mean by maximum? Is it possible that K=0?
> - Yes, $\mathcal{W}$ contains information up to K-th hop neighbor.
> - $K$ is always 1 in the experiments. We have made this more clear this in Sec. 3.
>
> ---
> Q5: How do you set the target vector z for the training data?
>
> - Since it is a classification problem, we simply use the one-hot vector as the target vector. We have updated Appendix C the clarify it.
>
> ---
> Q6: What is the motivation of the design of the feature transform layer? Why do we need to change the number of feature nodes? In the appendix F, why do you use the Flickr dataset for evaluating feature transform layers?
>
> - The feature transform layer defined in App F is able to further reduce the number of parameters if the number of feature nodes are reduced/changed layer by layer. (Although we shown in Sec. 5.4 that the featur broadcast layer requires the least number of parameters.)
> - Flickr is a very large dataset. For efficiency, we use the feature transform layer to further reduce the number of parameters.
> - We have updated the Appendix F to make this motivation more clear.
>
> ---
> Some minor issues:
>
> Q7: What is the motivation of using $\text{sgnroot}$ in Eq. (5)?
>
> - Inside the operator, it is $x\cdot y^T$, thus we use $\text{root}$ to get the number magnitude of $x$ and $y$.
> - We want to keep the sign of the correlation $x\cdot y^T$, but $\text{root}$ cannot get real values for negative inputs, thus we put the operator $\text{sign}$ outside of the $\text{root}$ operator.
> - The above insights inspired us to define the operator $\text{sgnroot}=\text{sign}(x) \sqrt(\left|x\right|)$.
>
> Q8: Some language errors and some statement are not precise and vague.
>
> - Thanks for your reminder, we have revised those statements in Sec 1.

---

> > ### Comment · AnonReviewer4 · 2020-11-25
> > **Need to further polish the paper**
> >
> > Thanks for the clarification. I am still have some words to say.
> >
> > Regarding Q1 (and also some other questions), making formuation general is good, but if you cannot make things clear, I would suggest removing some general settings. I just had a glance of the revised paper and found on page 6, below Eq.(9),  $C=1$ should be $c=1$. This is very like the using of $k$ and $K$. The notation confusion issues could make the paper hard to read. I feel the authors should re-examine every notation to make sure its meaning is clear.
> >
> > Regarding Q2, I can't understand edge weight=0. In this case, the edge simply does not exist. If you set edge weights to 1 or 0, it means no edge weight is acutally defined.
> >
> > Regarding Q3, I can't understand the word ``the weights are still associated with edges, while its availability in the continuum in Eq. (1) is associated with the nodes''. How do you associate edge weights to nodes? Why do this?
> >
> > Regarding Q4 (related to the issue of Q1), the use of $k$ and $K$ is still confusing (k=1 or K=1?).

---

### Official Review · AnonReviewer3 · 2020-10-27
**the paper presents a method by which the lifelong learning techniques developed for CNN can be applicable to GNN**

**Rating:** 6
**Confidence:** 4

**Review:**

The continuous learning on the graph neural network is restricted by the mechanism of the graph neural network itself. In order to enable the continuous learning approaches to be directly applied to GNN, the authors propose Feature Graph, which converts the node classification problem into graph classification by converting nodes into graphs. Therefore, we can apply current lifelong learning techniques to GNNs. Experimental results show that the proposed method overperforms the baselines. The main contribution of the paper is the idea of the converting technique, which makes it possible to employ existing methods to deal with the problem of continuous learning on the graph neural network. Overall, the paper is well written and is easy to follow.

Concerns:
1. Using the feature's cross-correlation matrix as the adjacency matrix of the neural network on the node sounds reasonable, but the validity of this idea lacks experimental explanation. In addition, how much do features contain the structure information of a graph? It is not discussed in this paper.

2. The dynamic increase of nodes is the main problem that limits the application of continuous learning approaches in graph neural networks. This problem still exists in Feature Graph. The author should discuss it.

3. The usage of task descriptors should be clarified, since their different usages may have a great impact on difficulty of problem

4. The experiments are not solid enough. Although Table 2 & 3 show positive result supporting proposed method, the experimental results are too brief to be convincing. Other results such as accuracy curves, or other baselines such as joint training should be provided. In addition, Cora, Citeseer and Pubmed are small graph datasets. Experiments on big graph datasets should be conducted.

Minor comments:
1. In Formula (9), the parentheses are redundant.

---

> ### Author Response · Authors · 2020-11-17
> **To Reviewer 3**
>
> Thanks very much for accepting our paper! We next address your concerns respectively.
>
> ---
> (1) Using the feature's cross-correlation matrix as the adjacency matrix of the neural network on the node sounds reasonable, but the validity of this idea lacks experimental explanation. In addition, how much do features contain the structure information of a graph?
> - We agree that it is very difficult to directly measure how much do features contain structural information. Therefore, we resort to emperical validation:
> 	- We first assume that the cross-correlation matrix is able to capture useful information for graph lifelong learning.
> 	- To verify this assumption, we first establish the model following the assumption and then show in Sec 5: with the cross-correlation matrix as the adjacency matrix, our feature graphs achieve much higher accuracy in lifelong graph learning than the method without taking the cross-correlatioin as adjacency matrix. These results successfully validate our assumption and provides the experimental explanation.
>
> ---
> (2) The dynamic increase of nodes is the main problem that limits the application of continuous learning approaches in graph neural networks. This problem still exists in Feature Graph. The author should discuss it.
>
> - The dynamic increase of nodes is converted into a continuum of feature graphs using our method.
> - The continuum is a basic setting of continual learning (Aljundi et al., 2019b), thus the problem of dynamic increase nodes can be solved/alleviated by applying the continual learining techniques.
> - The main challenge of continual learning from the continuum is the **forgetting issue**: the learned knowledge from earlier continuum may be easily forgotten since the earlier data cannot be available anymore.
> - We show in Sec. 5 that feature graphs achieve very low forgetting rate.
> - We have made this more clear in the new version (Sec 4.2).
> ---
> (3) The usage of task descriptors $t_i$ should be clarified, since their different usages may have a great impact on difficulty of problem.
>
> - The task decriptors $t_i$ in the experiments are integers numerating the tasks (at the end of Sec. 3 of initial version), which is equivalent to the class labels of samples $y_i$. In other words, we don't need $t_i$ during both training and testing (we only need sample labels $y_i$ during training).
> - However, we still define the task descriptors for compatibility with some continual learning techniques, which require such information.
> - We have made this more clear in the new version (Sec. 3).
>
> ---
> (4) Although Table 2 & 3 show positive result supporting proposed method, the experimental results are too brief to be convincing. Other results such as accuracy curves, or other baselines such as joint training should be provided. In addition, Cora, Citeseer and Pubmed are small graph datasets. Experiments on big graph datasets should be conducted.
> - We added a per-class precision figure and a per-class forgetting rate comparison figure in Fig 3 (a) and Fig. 3 (b) in the new version.
> - We also showed the training accuracy curves in Fig 2 (a) and Fig. 2 (b). A large dataset experiment, Flickr, was conducted in Appendix F.
> - Thanks for your suggestion, we will conduct more experiments in the future.

---

### Official Review · AnonReviewer2 · 2020-10-28
**Interesting problem, but some model designs need further clarifications**

**Rating:** 5
**Confidence:** 4

**Review:**

Summary:
This paper aims to bridge GNNs with life-long learning so that the catastrophic forgetting problem in graph-structured tasks is alleviated. Specifically, the major contribution seems to be transforming the original graph into a feature graph so that the node classification problem is transferred into a graph classification problem with isolated samples. Meanwhile, feature interactions are modeled in constructing edges of the feature graph. Experiments on three citation graphs demonstrate the effectiveness of the proposed method.

Pros:
(+) GNNs + lifelong learning seems to be a novel problem that has been seldomly studied.
(+) The proposed framework can be applied to different GNNs.
(+) The paper is overall well written and easy to follow.

Negative points are as follows:
(1) Converting the original graph into a feature graph is not adequately justified and may have severe drawbacks. The main motivation for such a conversion is to transform connected nodes in the original graph into isolated samples so that the existing lifelong learning methods can be applied. However, the whole point of using GNNs is to pass and exchange messages between different nodes. If each node is regarded as an isolated sample, the relationships between nodes are completed ignored (except the feature co-occurrence statistics). In other words, the resulted model is essentially feature-centric and basically does not preserve any structural information in the original graph. For example, it is highly likely that the proposed model cannot preserve motifs or structural roles of nodes, nor to handle structural-driven tasks such as link prediction. The authors need to further clarify this major model design.
(2) Following how to convert nodes into isolated samples in GNNs, a well-known method is to regard each node as an ego-graph, since the representation of a node in GNNs only depends on its k-hop neighbors (see GraphSAGE and [1-2]). Such a conversion will also transform the node classification problem into a graph classification problem, similar to the paper’s arguments, but without losing structural information. Thus I am wondering whether or why such a method cannot be directly applied in the lifelong setting (from Section 3, the k-hop neighbors’ information should be available).
(3) In experiments, the authors adopt three citations graphs. Though I acknowledge they were commonly used as benchmarks in GNNs, recent studies suggest these small datasets may be not adequate in comparing different methods [3-5]. Thus, more experiments on larger datasets may be needed. (The experimental results on the Flickr dataset in the Appendix are puzzling since the results show that not using memory outperforms using memory, indicating that graph lifelong learning may even not be a proper setting.)
(4) There are also a few missing related works [6-7].

Minor:
(1)	In related works, JK-Net and DiffPool are GNN architectures (proposing jumping connections and differentiable pooling) rather than new sampling techniques.
(2)	It should be noted that building feature graphs and considering feature interactions in GNNs have also been studied recently, see [8-10]. But since they are informal publications or very recent w.r.t. the submission, I only suggest the authors compare them in an updated version and do not consider this as a negative point.

[1] Link Prediction Based on Graph Neural Networks, NeurIPS 2018
[2] Graph Meta Learning via Local Subgraphs, arXiv:2006.07889
[3] Pitfalls of Graph Neural Network Evaluation, arXiv:1811.05868
[4] Open Graph Benchmark: Datasets for Machine Learning on Graphs, arXiv: 2005.00687.
[5] Benchmarking Graph Neural Networks, arXiv: 2003.00982
[6] Lifelong representation learning in dynamic attributed networks, Neurocomputing 2019.
[7] Streaming Graph Neural Networks via Continual Learning, CIKM 2019.
[8] Cross-GCN: Enhancing Graph Convolutional Network with k-Order Feature Interactions, arXiv:2003.02587.
[9] CatGCN: Graph Convolutional Networks with Categorical Node Features, arXiv:2009.05303.
[10] AM-GCN: Adaptive Multi-channel Graph Convolutional Networks, KDD 2020.

Based on the above comments, I am currently leaning towards rejection. I am happy to improve my scores if the authors can further justify their proposed method.

---

> ### Author Response · Authors · 2020-11-17
> **To Reviewer 2**
>
> Thank you for agreeing that “GNNs + lifelong learning seems to be **a novel problem** that has been seldomly studied", "The proposed framework **can be applied to different GNNs**." and "The paper is **overall well written and easy to follow**." We next address your concerns respectively.
>
> ---
> (1) This method does not preserve any structural information in the original graph.
> - This is not true. Feature graph encodes the structural information of the original graph into the feature adjacence matrix $A_{k,c}^{\mathcal{F}}$ as defined in Eq. (5).
> $$A_{k,c}^{\mathcal{F}} (x) \triangleq \operatorname{sgnroot} \left( \mathbb{E} \left[w_{x,y}x_{[:,c]} y_{[:,c]}^{\rm {T}} \right] \right), \quad \forall y \in N_k(x)$$
> where $k$ is the k-hop neighbors of the original graph. In this setting, the structural information is encoded by the cross-correlation with neighbors $\mathbf{y}$, meaning that **the structural information is retained in another form**.
>
> ---
> (2) Methods like GraphSAGE will not lose structural information and I am wondering whether or why such a method cannot be directly applied in the lifelong setting.
> - As answered in the first question, we don't lose the structural information.
> - As mentioned in 2nd paragraph of Sec 5, GraphSAGE needs to traverse the entire graph for every layer which is impossible for lifelong learning, but it is possible to be applied to the lifelong learning via slight modification (see more details in Sec 5). However, its performance is much lower than our method. This was what we showed in Sec 5:
> 	- In Table 2, the feature graph achieves 5.5%, 1.9%, and 5.6% higher accuracy than GraphSAGE for data-incremental learning tasks.
> 	- In Table 3, we achieves an average of an average of 5.8% higher accuracy than GraphSAGE for class incremental tasks.
> 	- Table 4 shows that our method only requires $\frac{1}{10}$ parameters of GraphSAGE.
>
> ---
> (3) More datasets test excluding Cora, Citeseer, and Pubmed are needed, since the Flickr dataset test in Appendix is puzzling: the results show that not using memory outperforms using memory.
>
> - We explained this phenomenon in **Page 14**: the overfitting effect. This is because we adopt a simple memory reply strategy following (Aljundi et al., 2019b): replaying the memory samples (updating the parameters by the memory samples) after each time we update parameters by new samples from the continuum, thus a larger memory or dataset leads to more learning steps and easier overfitting.
> - We could overcome this phenomenon, for example, by updating parameters from memory after receiving **more** samples in the continuum. We believe that a better-designed updating strategy is able to further improve the performance, but it is out the scope of this paper, as we only focus on the feature graphs.
> - In **Sec F**: Even a non-continual learning method GraphSIANT achieves only an upper bound accuracy of 0.51 on **Flickr**, while we achieve an average accuracy of 0.470 and 0.468 for the data-incremental and class-incremental tasks, respectively. **The performance on Flickr verifies the effectiveness of our method.**
>
> ---
> (4-6) Missing related works [6,7] and other minor issues.
> - We have added the related work [6] in the new version and find [7] is published in CIKM 2020 (2020 Oct, arXiv is 2020 Sep 23rd).
> - Thanks for your suggestions and we have corrected the context of JK-Net and DiffPool in the new version.

---

> > ### Comment · AnonReviewer2 · 2020-11-20
> > **Thanks for Responses and Further Explaining My Concerns**
> >
> > I thank the authors for their detailed responses, but maybe I was not entirely clear regarding weakness (1)-(3). Now let me further explain my concerns.
> >
> > (1) I acknowledge that the feature co-occurrence statistics within k-step neighbors, which is one kind of structural information, is preserved and such information may allow the model to handle node classification on some datasets, e.g., Cora, CiteSeer, PubMed. However, I am wondering whether it is sufficient, e.g., for other tasks (e.g., link prediction) or other datasets (where node features are less smooth and dominant, e.g., see [1] for quantitative analysis on feature smoothness of different datasets).  Without further proving the rationale of this design, the applicability of the proposed method is limited. This is also one reason I asked for more experimental evidence in (3).  It seems that this concern is also raised by two other reviewers.
> > [1] Measuring and Improving the Use of Graph Information in Graph Neural Networks, ICLR 2020
> >
> > (2) I only cite GraphSAGE because it explicitly introduces the concept of k-step ego-graph, but do not mean only GraphSAGE can be applied in the authors' scenario. For example, the authors claim GCN, GAT "require a full adjacency matrix", which is not entirely true, i.e., if you use the matrix version of GCN to calculate the entire embedding matrix, you will need the full adjacency matrix, but if we only calculate the embedding of node v_i and let's say we use 2 layers in GCN, then only edges/node features within the two hop-neighbors of node v_i are needed. Thus, I don't think only comparing with GraphSAGE is sufficient.
> >
> > (3) It's okay if the proposed method only overfits in Flickr, but I am wondering whether this implies that the proposed method is likely to overfit in any large-scale graph (since no other large-scale graph is adopted). Besides, comparing the absolute performance of GraphSAINT and the proposed method is not very convincing since there is no reference line. Thus, though I acknowledge that the current experiments show some evidence, I am suggesting adopting more datasets will make the proposed method more convincing.

---

> > > ### Author Response · Authors · 2020-11-24
> > > **To Reviewer 2**
> > >
> > > Q1: I acknowledge the structural information is retained in the feature graphs, however this method may not be sufficient for other tasks, e.g. link prediction, or data, where node features are less smooth [1], hence we need more experimental evidence as mentioned in Q3.
> > >
> > > - Thank you very much for agreeing us with this contribution! It seems that R2 poses new questions on *(a) other graph tasks* and *(b) feature smoothness* (defined in [1]). Here is our insight:
> > > - Other Graph Tasks:
> > > 	- The contribution of this paper is to bridge GNNs with life-long learning so that the catastrophic forgetting problem in graph-structured tasks is alleviated. Since it is a first try, we do **not** aim to demonstrate that it is suitable for all graph tasks. Instead, we only focus on **node classification** and would like to do tasks like link prediction in the future.
> > > - Our method **has no conflicts** with [1], which uses the metric, feature smoothness, to construct a graph model. [1] can also be applied to our feature graphs. This is because
> > > 	- A feature node is a feature of a regular node, thus feature graph retains the original feature elements/numbers. Therefore, it does **not** change the feature smoothness from the perspective of a regular graph (at least the first layer, as the nodes features are transformed by different learnable parameters in following layers).
> > > 	- Feature graph is established via new edges definition (feature adjacency matrix), meaning that it is a new graph **topology** (not only a graph model). Therefore, as  mentioned in Sec 4.3, any existing graph models, **including [1]**, can be applied to feature graphs.
> > > 	- For example, a simple way is to introduce the smoothness metric to the feature adjacency matrix in Eq (5).
> > > $$A_{k,c}^{\mathcal{F}} (x) \triangleq \text{sgnroot} \left( \mathbb{E} \left[q_{x,y}^{k} \cdot x_{[:,c]} y_{[:,c]}^{\rm {T}} \right] \right), \quad \forall y \in N_k(x)$$
> > > where definition of $q^{k}_{x,y}$ can be similar to Eq (2) in [1], meaning that neighbors can contribute larger information with larger feature smoothness. We also mentioned that some other methods, such as graph attenion, kervolution, can also be applied to feature graphs (Sec 4.3). However, they are out of the scope of this paper, as we only aim to demonstrate the graph topology, feature graph, is effective in lifelong graph leanring. Thanks for your suggestions, we would like to try the feature smoothness method using feature graph when feature smoothness is the bottleneck for lifelong graph learning.
> > > ---
> > > Q2: GCN models can also be directly used for lifelong learning if we only calculate the embedding of node using 2 layers GCN.
> > >   - The graph lifelong learning will be meaningful **only** when the number of graph layers $L$ is larger than the availability of $K$-hop neighbors, i.e., $L>K$, otherwise it will degrade to one-epoch training of traditional models. Note that $L>K$ is also the setting in our experiments, and your setting ($L=K=2$) does not apply to our scenario.
> > >    - GCN **cannot** be applied to our setting of lifelong graph learning because the $l$-th ($K<l\leq L$) layer node embeding cannot be computed if we only traverse the continuum once, which is the setting of lifelong learning. (Meaning that it still quires a full adjacency matrix.)
> > >    - Thanks for your reminder, we have made this more clear in the new version.
> > > ---
> > > Q3: More datasets will make the proposed method more convincing.
> > >
> > > - Thanks for your suggestions, we have test our feature graph on a much larger graph dataset, ogbn-arxiv, and compare with GraphSAGE. As a comparison, we also list the statistics of Flickr dataset.
> > >   |	Datasets  | Nodes  | Edges  |  Classes | Features  | Label Rate  |
> > >   |---|---|---|---|---|---|
> > >   | ogbn-arxiv |169,343  | 1,166,243  | 40  | 128  |  0.5 |
> > >   | Flickr | 89,250| 899,756  |  7 | 500  | 0.5 |
> > > - Setting: we use the memory size of 500 and update the model for 5 times using each newly available samples. During training, we adopt the Adam optimization method, where a mini-batch size of 64 and a learning rate of 0.01 are adopted.
> > >
> > > - Feature graph achieve a lower performance in the non-lifelong setting, while a higher performance in lifelong learning. This further demonstrates that the feature graphs have much lower forgetting rates in lifelong learning.
> > >   |	Tasks  | Non-lifelong  | Lifelong |
> > >   |---|---|---|
> > >   | GraphSAGE | 0.615 | 0.532 |
> > >   | Feature Graph (ours) |  0.595 | **0.550**

---

> > > > ### Comment · AnonReviewer2 · 2020-11-24
> > > > **Responses**
> > > >
> > > > I thank the authors for their responses and appreciate the new results on ogbn-arxiv. However, while the paper has become clearer, some concerns regarding the contribution and scope of this paper also arise. Specifically,
> > > >
> > > > (1) Since the authors have agreed that their method currently only deals with node classification, the applicability of the proposed method is limited than it seems to claim (or at least than I understood in the first place), i.e., GNN + life-long learning vs. GNN + life-long learning for node classification.
> > > >
> > > > (2) Now I understand that under the specific settings in this paper (L > K), other GNNs cannot be applied. However, such lifelong learning settings seem rather deliberately designed, i.e., we have to use a deep GNN while having small neighborhood information (but we also cannot have no neighborhood at all). I am not sure when this setting will become practical or why couldn't we simply reduce L to match K.
> > > >
> > > > (3) Though the authors have shown their proposed method can beat GraphSAGE in the lifelong setting, the feature graph has a considerable margin compared to GraphSAGE in the non-lifelong setting, which seems to verify my suspicion that the feature graph will lose structural information.

---

> > > > > ### Author Response · Authors · 2020-11-24
> > > > > **To Reviewer 2**
> > > > >
> > > > > Thanks for your quick reply and thanks for agreeing that our method has no issues with the feature smoothness.
> > > > >
> > > > > Q1: We believe that only focusing on one application (node classification) is not an issue. Many great papers, including GCN, GAT, GraphSAGE, etc, also only focus on node classification. It might be too aggressive to focus on too many tasks in one paper.
> > > > >
> > > > > Q2: Potential applications and why couldn't we simply reduce $L$ to match $K$.
> > > > > - Thanks for agreeing that GCN is not able to be applied to the scenario.
> > > > > - It has been shown in many papers that deeper GNN (larger $L$) is able provide higher performance. One of the newest is [a]. However, in practise, we normally couldn't control the maximum available $K$-hop neighbors, hence developing an lifelong graph learning algorithm that does't not require $L\leq K$ will be very useful.
> > > > > - Moreover, recent deep learning (DL) techniques require intensive memory computing resources, thus lifelong/continual learning for DL would be necessary for pratical AI, hence the settings of  lifelong graph learning in this paper is also very important.
> > > > >
> > > > > [a] Fast and Deep Graph Neural Networks, AAAI 2020.
> > > > >
> > > > > Q3: The performance in non-lifelong setting seems to verify my suspicion that the feature graph will lose structural information
> > > > >
> > > > > - Thanks for agreeing that the proposed method can beat GraphSAGE in the lifelong setting.
> > > > > - It is **not sufficient** to conclude that " feature graph will lose structural information" only based on the "performance in non-lifelong learning". As we mentioned before, there is no standard way to measure how much the structural information is retained. As you agreed before, our method is able to retain a kind of structural information, but mearsuring how much information is retained is not the scope of this paper.
> > > > > - Actually there are many potential explanations for "performance in non-lifelong learning". For example, it may because the feature graph use only 1-hop neighbors in the non-lifelong setting (follow the same model settings in lifelong learning), while GraphSAGE uses maximum 2-hop neighbors in non-lifelong learning. But this is not what we focus.
> > > > > - We believe that a slight lower performance in non-lifelong learning is not an issue, as we aim to demonstrate the proposed method is effective for **lifelong graph learning**. Moreover, a lower performance in non-lifelong learning but higher performance in lifelong learning further validates that the feature graph has much lower forgetting rates.
> > > > >
> > > > > Last but not least, we sincerely thank R2 for a large number of suggestions and comments. They are really helpful for our future works.

---

> > > > > > ### Comment · AnonReviewer2 · 2020-11-25
> > > > > > **Thanks for Clarifications**
> > > > > >
> > > > > > I thank the authors for clarifications, but I still think the scope and contribution of the paper in the current form is somewhat limited (at least than I presumed in my initial reviews). Thus I decide to keep my score. Nevertheless, I do think the clarity of the paper has improved after the revisions, and appreciate the authors' efforts in the discussion phase.

---

### Official Review · AnonReviewer1 · 2020-10-28
**Interesting idea but theoretical justification and experiments are not convincing**

**Rating:** 5
**Confidence:** 4

**Review:**

This paper aims to solve the problem of lifelong graph learning. Thus far, the topic about graph learning and lifelong learning is still underexplored. This paper proposes a new graph topology based on feature interaction, which takes the features as nodes and turns the nodes into graphs, and thus formulates a regular lifelong learning problem by defining the feature graph continuum. The authors conduct experiments on three popular citation graph datasets including Cora, Citeseer, and Pubmed.

Pros:
1. This paper presents a novel strategy to transform the regular graph to a feature graph. This converts the original problem of node classification to graph classification, where the increasing nodes are turned into training samples. It makes the graph learning applicable to the continual learning.

Cons:
1. In Section 1, the authors mention that--“It takes the features as nodes and turns the nodes into graphs. This converts the problem of node classification to graph classification. In this way, the increasing nodes become training samples”. In the lifelong learning, this strategy will not only increase the node samples but also the edges between new and old nodes. Although the feature graph continuum and random sample rehearsal strategy are proposed, scalability might still be a concern.

2. It is unclear whether the proposed feature graph and feature adjacency matrix could effectively capture useful information from neighbors. More justifications and theoretical analysis shall be provided. In addition, evaluating the performance of feature graph in some traditional graph learning tasks would be helpful.

3. In the experiments, the authors only compare their method with the modified GraphSAGE method. Although this topic about the combination of graph learning and continual learning is relatively new, there exists several relevant papers such as Continual Graph Learning (March, 2020). It would be more convincing if the authors could conduct comparisons with other graph continual learning or lifelong learning methods.

---

> ### Author Response · Authors · 2020-11-17
> **To Reviewer 1**
>
> Thank you for agreeing that “this paper presents **a novel strategy** to transform the regular graph to a feature graph” and “**It makes the graph learning applicable to the continual learning**.” We next address your concerns respectively.
>
> ---
> (1) This strategy will not only increase the node samples but also the edges between new and old nodes. Although the feature graph continuum and random sample rehearsal strategy are proposed, scalability might still be a concern.
>
> - Noted that each regular node $x\in R^{F\times C}$ is associated to $F$ feature nodes $x_i\in R^C$. This means that a feature node is just one feature of a regular node. Therefore, our strategy does **not** increase the total number of data element $E=FC$.
> - Feature graphs have **constant** ($F\times (F-1)/2$) number of edges  and we  provided complexity analysis in **Sec 4.4**: feature graph roughly has complexity of $O(E^2+nF^2)$, where $n$ is the expected number of neighbors in the continuum. Note that they are irrelevant to the task number, thus it has **constant complexity** with the increased learning tasks/samples.
> - We also provided model size comparison in **Sec 5.4**, which shown that our method only requires $\frac{1}{10}$ parameters, which improves the scalability.
> - The rehearsal strategy we used has **constant memory consumption** (fixed number of samples) and we show in **Sec 5** that it reduces the forgetting issue by a large margin.
> - Our strategy converts the increasing samples into a continuum, which is a basic setting of continual learning, in which the forgetting issue rather than the scalability is the challenge. Other continual learning methods that do not require memory are also applicable to our methods.
>
> ---
> (2) It is unclear whether the proposed feature graph and feature adjacency matrix could effectively capture useful information from neighbors.  More justifications and theoretical analysis shall be provided. Evaluating the performance of feature graph in some traditional graph learning tasks would be helpful.
> - We used the cross-correlation matrix to capture the relationship from neighbors. To demonstarte its **effectiveness**, we first assume that it can effectively capture the useful information by taking it as an adjacency matrix, we then established the model and **verified this assumption** by showing that this structure achieves an average of **5% higher** accuracy in graph continual learning (Sec 5).
> - We agree it is still an open challenge to theoretically evaluate whether a method could effectively capture useful information, and this challenge exists for many other machine learning algorithms (Emperical justifications are widely used, which is also what we provided). It is a very interesting topic and we would like to conduct it in the future.
> - As in the **third paragraph of Sec 5**: we achieved a little bit higher performance than GraphSage in the **traditional graph learning tasks**, but much higher performance in continual graph learning. This further demonstrated the effectiveness of our method in continual learning.
>
> ---
> (3) It would be more convincing if the authors could conduct comparisons with other graph continual learning or lifelong learning methods, such as Continual Graph Learning (March, 2020).
> - Thanks for your suggetion, but "Continual Graph Learning (**CGN**)" is an arXiv paper and not published yet. It has a very different formulation and tested several exemplar selection methods (**Sec 2.1**). Moreover, it doesn't provide implementation details, which makes it difficult to compare.
> - To the best our knowledge, feature graph is one of the first methods that make continual learning techniques applicable to graph learning.

---

### Decision · Program_Chairs · 2021-01-07
**Final Decision**

**Decision:**

Reject

**Comment:**

The reviewers initially assessed this paper as slightly below the acceptance threshold. The reviewers seem to agree on the novelty and potential impact of this project, but they also highlighted the lack of clarity of the manuscript including lack of clarity in the method used to encode the graph data.

As the authors noted, graph-related questions were the focus of most of the comments and questions from the reviewers. This is not because the reviewers did not understand and assess the method from the continual-learning side (I am also meta-reviewing several continual-learning papers and I believe that I can assess the novelty of this work). As I wrote above, reviewers were convinced of the paper's motivation.

The authors provided good responses and discussed with at least one reviewer thoroughly. These interactions seem to have clarified important aspects of your proposed methodology and notably the properties of your graph-construction method. I found that your new results on larger datasets also provide an improvement. However, to be properly assessed, this number of clarifications regarding the core method requires a new round of reviews. The discussions have also highlighted some of the limits of your approach which do not seem to be acknowledged in your paper. This includes the discussion with reviewer2 regarding constraints on L & K, node classification (also I find that one to less important), and comparison to GraphSage on the non-lifelong learning scenario.

Overall, and while I agree that continual learning from graph data is an important and unexplored problem, I also find that the current manuscript lacks clarity and, even though the ICLR discussion allowed reviewers to discuss these with the authors, there are still significant ways to improve the clarity of the current manuscript. As a result, I do not recommend acceptance of the current manuscript.

I strongly suggest the authors keep on working on their manuscript as their idea seems to have potential and I would imagine that it may become one of the first works in a new interesting line of research.